# Optimizing Image Enhancement: Feature Engineering for Improved Classification in AI-Assisted Artificial Retinas

**DOI:** 10.3390/s24092678

**Published:** 2024-04-23

**Authors:** Asif Mehmood, Jungbeom Ko, Hyunchul Kim, Jungsuk Kim

**Affiliations:** 1Department of Biomedical Engineering, College of IT Convergence, Gachon University, 1342 Seongnamdaero, Sujeong-gu, Seongnam-si 13120, Republic of Korea; asif@gachon.ac.kr; 2Department of Health Sciences and Technology, Gachon Advanced Institute for Health Sciences and Technology (GAIHST), Gachon University, Incheon 21936, Republic of Korea; rhwndqja@bme.gachon.ac.kr; 3School of Information, University of California, 102 South Hall 4600, Berkeley, CA 94720, USA; hyunchul@berkeley.edu; 4Research and Development Laboratory, Cellico Company, Seongnam-si 13449, Republic of Korea

**Keywords:** classification, deep neural network, image processing, artificial intelligence, artificial retina, AI-enabled sensors, smart sensors

## Abstract

Artificial retinas have revolutionized the lives of many blind people by enabling their ability to perceive vision via an implanted chip. Despite significant advancements, there are some limitations that cannot be ignored. Presenting all objects captured in a scene makes their identification difficult. Addressing this limitation is necessary because the artificial retina can utilize a very limited number of pixels to represent vision information. This problem in a multi-object scenario can be mitigated by enhancing images such that only the major objects are considered to be shown in vision. Although simple techniques like edge detection are used, they fall short in representing identifiable objects in complex scenarios, suggesting the idea of integrating primary object edges. To support this idea, the proposed classification model aims at identifying the primary objects based on a suggested set of selective features. The proposed classification model can then be equipped into the artificial retina system for filtering multiple primary objects to enhance vision. The suitability of handling multi-objects enables the system to cope with real-world complex scenarios. The proposed classification model is based on a multi-label deep neural network, specifically designed to leverage from the selective feature set. Initially, the enhanced images proposed in this research are compared with the ones that utilize an edge detection technique for single, dual, and multi-object images. These enhancements are also verified through an intensity profile analysis. Subsequently, the proposed classification model’s performance is evaluated to show the significance of utilizing the suggested features. This includes evaluating the model’s ability to correctly classify the top five, four, three, two, and one object(s), with respective accuracies of up to 84.8%, 85.2%, 86.8%, 91.8%, and 96.4%. Several comparisons such as training/validation loss and accuracies, precision, recall, specificity, and area under a curve indicate reliable results. Based on the overall evaluation of this study, it is concluded that using the suggested set of selective features not only improves the classification model’s performance, but aligns with the specific problem to address the challenge of correctly identifying objects in multi-object scenarios. Therefore, the proposed classification model designed on the basis of selective features is considered to be a very useful tool in supporting the idea of optimizing image enhancement.

## 1. Introduction

It has been almost a decade since artificial retinas [1] came out to make vision possible for the blind in the healthcare industry [2,3,4]. The development and advancement of artificial retinas, also known as retinal prostheses or bionic eyes, have made significant strides in allowing visually impaired individuals to regain some form of vision. The concept of an artificial retina [5], which involves implanting a light-sensitive device in the eye to directly stimulate retinal cells, has revolutionized the field of vision restoration and rehabilitation for the blind [6,7].

Figure 1 represents the components and processes involved in AI-based artificial retina support [8]. Generally, the first step is to capture the scene using a camera view. Secondly, processing is conducted for various reasons to prepare the image signal for transmission to the chip. This process involves the use of an AI model, which classifies the objects as primary and non-primary. It can play a role in solving a challenge or implementing optimization [9]. The third process is the transmission of the image/signal to the brain via the optic nerve, which helps restore vision in blind people.

The artificial retina processing flow demonstrates that an AI-assisted model, built based on machine learning (ML) [10] or deep learning (DL) [11], can be incorporated for image enhancement, which can play an important role in overcoming the challenges/limitations of a retina chip that affects vision.

There are many challenges to be addressed in the future development of artificial retinas [12,13]. Some of the challenges that can be addressed using artificial intelligence (AI) [14] are improving image resolution and clarity, filtering out unnecessary objects, enhancing visual perception to overcome the limitation of an artificial retina’s resolution, and developing better integration of image processing algorithms to mimic the functionality of a human retina. Overall, the primary challenge is to produce such an enhanced image such that not only are the unnecessary objects removed, but vision is not compromised. This challenge opens new directions and research challenges, e.g., the identification of multiple objects in complex scenes within a grayscale image with a limited pixel resolution.

The integration of AI in artificial retina development has revolutionized the field and paved the way for more advanced and efficient vision. The use of AI in artificial retina technology has many applications, including image enhancement algorithms [15], object-recognition and tracking [16], and adaptive learning systems [17]. Furthermore, AI can also play a crucial role in addressing challenges and limitations in artificial retinas by analyzing large amounts of data, identifying patterns, classifying objects in the image, and filtering the unneeded objects for the optimization of image enhancement. Overall, the use cases of AI hold great potential in addressing the challenges faced in improving technology.

The existing techniques used for image enhancement primarily focus on capturing the edge [18] and background [19] information. Although the image is enhanced, the results are promising for images having multiple objects [20,21]. This makes it difficult for the artificial retinal patients to identify such objects. The objective of this work is to address the challenge of improving the image in such a way that the patient finds it convenient to identify the objects. To this end, the proposal aims primarily focus on only showing the primary objects, which results in an improved image. This demands the need of a classification model for classifying the primary objects. In this work, feature engineering is performed for finding the attributes associated with objects that play an important role in identifying the object as primary or non-primary. The main features used in this work are objects, their depths, size, location, and bound boxes. Based on these features, a deep neural network (DNN) is modeled, built, and trained for classifying the multiple objects found in the image as primary or non-primary. DNN is widely used in the field of image processing for automating the features’ space compared to the traditional classifiers [22,23].

Image-enhancement [24,25] is necessary due to the following limitations. Firstly, the chip can express very few pixels, which means that all information cannot be represented accurately. As a result of this, only 2D grayscale and low-resolution image signals are allowed to be used for transmission, which makes it insufficient for representing real objects. Secondly, there is no color support, which forces the chip to represent the image signal in grayscale only. Third, the spike frequency is very slow, so the image must be in high grayscale contrast to optimize the image enhancement. All of the above-mentioned limitations conclude that these eye implants cannot convey much information to the brain, and the hardware of the chips are not capable of transmitting enough information to the brain, thus urging the need of a mechanism that may assist somehow [26].

Edge detection is a widely used technique to enhance images for producing easily identifiable objects. Although the boundaries of entities in an image can aid in improving vision with a limit pixel resolution, the multiple-object scenarios are overlooked and not addressed effectively. To address the challenge of multiple objects, several techniques are utilized, such as multi-focus image fusion [27].

The integration of feature engineering and DL play an important role in correctly classifying the objects as primary or non-primary [28,29]. The commonly used features are object size, area it covers, and its distance. Altogether, these features play a vital role in defining the objects as important. Using these rich features significantly improves the model’s performance, which needs to be tailored to solve a particular problem. Overall, the role of selective features cannot be neglected, as it is a key factor that directly influences any model’s performance. For this particular study, the classification model is opted to leverage these features such that it can accurately classify the top five, four, three, two, and one object(s) as primary or non-primary.

Therefore, as an attempt to enhance the image, an AI-assisted model is proposed to be responsible for classifying the objects in an image as primary or non-primary, based on whether the objects are important to show or not. Taking this classification [30] as a base, non-primary objects could be removed from the image, and the image could be scaled to an optimized resolution such that the image is enhanced and is transmittable to the chip with the need for an optimized image enhancement requirement. This supports the idea of integration of the classification model into an artificial retina system for the purpose of enhancing vision and improving its usability in real-world scenarios.

### 1.1. Related Work

Processing in an artificial retina is a burdensome challenge, and little research has been conducted to overcome its limitations. Due to limited hardware capability, ML and NN can be leveraged to address this challenge [31]. Recent advances in this field have developed and resulted in many systems adopting such models to overcome processing and computational challenges. Much of the focus in this area is primarily on improving hardware, but not much emphasis has been placed on the software side [32], where AI models can help enhance images before they are transmitted to the brain.

Image processing techniques [2,33] have widely been used across several domains in the healthcare industry. These techniques are revolutionizing different sectors, for example, medical imaging, trouble shooting, disease diagnosis, and surgical guidance. Some key applications of image processing techniques in healthcare are image enhancement, segmentation, and registration. Each of the techniques have different roles in vision [34], with segmentation being the technique to define segments in vision, and enhancement to improve its quality; both can be leveraged to support the idea of optimizing image enhancement by the use of feature engineering.

The edge detection [18] and simulated phosphene [35] of an image is shown in Figure 2. The edges do not provide a clearer view due to the presence of irrelevant objects and background information. This result leads to difficulty in the identification of objects in vision. On the right side, the simulated phosphene is shown, where it can be analyzed that the objects are not easily identified as well, thus urging the need to further enhance the vision such that multiple objects can be easily identified. This will enable systems like an artificial retina to address the challenges of representing real-world objects in a better way, with its current limitations imposed by its hardware.

Image segmentation can also be seen as an application of image processing in healthcare [36]. As it allows the definition of segments involved in vision, this can support the idea of object filtering based on primary objects. The overall practicality of this study’s application is the integration of a primary object classification model and object filtering. This can be implemented by the use of recurrent refinement networks (RRN) [37] to extract the masks of segments in vision. This enables filtering the objects that are primary, hence removing the unnecessary information from the view, thus enhancing vision.

Image fusion techniques can also be utilized to improve the input images. Multi-focus image fusion (MFIF) is one such technique in which two or more images are aggregated, where the focus of the image can be identified as a primary area in vision. This is one possible way to leverage image fusion. In contrast, our research focuses on defining the primary objects based on features such as object size, area it covers, and its distance from the viewer. All these provide support in defining the object’s importance.

Image enhancement can make it significantly easier to identify multiple objects in an image with limited resolution available, if done correctly. Images must be pre-processed in to enhance them with the appropriate color contrasts. An additional perspective can be added for further enhancement by removing unnecessary objects from the images that need to be transmitted to the retinal chips. Image enhancement is very crucial, as the cheaper retinal chips have a slow image processing rate, and good retinal chips are very expensive. Therefore, a possible solution is to remove unnecessary objects from the image and apply filters over the primary objects to improve the visualization experience of blind people.

Due to the limitation of spike frequency, the visualization experience is slowed down and can be frustrating for the patient. To address this challenge, several classification [30,38,39] models can be used in the artificial retina’s video-processing unit, which assists in the necessary filtering by specifying which objects are primary and non-primary. Therefore, by taking advantage of this, extra pixels can be removed and better color contrasts can be applied to make multiple objects easily identifiable. This not only improves the experience, but also reduces the risk of eye tissue damage. Overall, this approach to the challenge is simple and effective.

When it comes to image processing, various techniques [40] are used for the classification problem. In various techniques specifically for primary object classification, the features are extracted. One such approach is the weighted feature fusion [27] of a convolutional neural network and graph attention network (WF-CNN-GAT), which transforms the image by the utilization of spectral features. In this work, several features are extracted from the images, which are objects, their depths, size, location, and bound boxes. These features are annotated to the dataset images. Furthermore, in [41], the authors employed a long short-term memory (LSTM) for posture recognition.

To solve the problem of the classification of primary and non-primary depends on several factors, which are discussed later in Section 5. It is important to mention here that a human being’s focus is developed based on the preceding events and many other reasons, stimulating our attention and focusing on a particular object. Focus is developed either by the movement in surroundings, object’s size, color, distance, etc. Considering the basic and possible annotations that can be derived by data provided by SSD [42], a depth-estimation model, and preprocessing the output there to derive the object sizes and distances, the classification model’s network has been designed, developed, and evaluated for assisting an artificial retina.

In binary classification, the criteria of evaluation for a multi-label classification is an important aspect to consider for a fair analysis [43]. Best practices [44,45,46] are adopted and applied on the benchmark datasets [47]. One of the best practices that we adopted is to consider the top five, four, three, two, and one object’s evaluation separately. This approach highlights the evaluation metrics fairly for all objects in an image, referring to the fact that object classification is fairly treated, i.e., a justified evaluation.

The aim of this research is to understand the importance of the application of object classification in the domain of artificial retinas. Adapting the chip to include software that can intelligently help filter out unnecessary objects will significantly improve its processing. To address this challenge, popular architecture designs are considered and evaluated based on a fair analysis criterion, i.e., top ‘n’ object analysis. Overall, the proposed changes to the system innovate and improve the visual experience of blind people.

### 1.2. Methodology and Major Contributions

The primary objective of this work is to choose specific features to be used in the primary/non-primary object classification problem. For this, the extraction of features was conducted following several steps. Firstly, annotation of the images was required, which involved finding objects and their associated features using various methodologies [42,48]. The top five objects from the images were extracted using a well-known model, i.e., single shot detector (SSD) [42]. As the model only gives us the classes of each object with their respective bound boxes and prediction scores, other features need to be derived from them. Secondly, depth estimations of the images are obtained using a PyTorch-based monocular depth estimation model trained on the dataset vinvino02/glpn−nyu [48].

Based on the image depth estimations, objects classes, and object bounding boxes, further features such as object area and depth can be derived. Since these two are the most important features, they need to be calculated accurately. A basic approach to calculating the area of an object can be conducted using bounding boxes, and calculating the depth of an object can be conducted using average depth intensities inside the bounding boxes. These approaches are erroneous and significantly affect performance. To calculate the area accurately, the depth of an object is required. Therefore, a mechanism has been developed to accurately estimate the depth of an object. Then, the actual size of each object is derived based on their depths. This is needed because the object covering the greatest part of the image may not actually be the primary object. As a result of this, each object was fairly treated and virtually [49,50] transferred to a common depth, and then the area and their actual size were calculated.

The final task in dataset preparation is to label the image data. As five objects are considered in this experimentation, five labels are required to be tagged as target variables of the dataset. Primary and non-primary labels are decided based on the actual size of an object and their respective depths.

The proposal also involves modeling a neural network (NN) [51] model to classify primary and non-primary objects in an image. For modeling an efficient model [52], a good network and premeditated features [53] are required. Regarding feature design, metrics affecting model performance need to be included in the training. The list of metrics used as input to the model includes a grayscale image and details of the top five objects, including their class, bounding box, area, depth, and score. Based on the analysis, these metrics are considered to play a vital role in deciding whether the object is primary or not. Also, the output of the model denotes a binary classification that needs to be annotated.

Since the dataset preparation part is complete, the model’s network is designed in such a way that its performance is not adversely affected. Inspired by CNN architecture [54,55], the first part of the model applies four 2D convolutions to the image and four 1D convolutions to the object detail matrix to extract initial features. Then, global 2D and 1D average pooling is applied to each branch, respectively. Both the initial features of the image and object details are rearranged and concatenated, followed by a dropout method. The earlier process gives us initial features combined together from the image and objects, which are fed to the linear layer. This gives us an output in the form of five binary outputs that refer to the primariness of each object.

The major list of contributions of this work are as follows:**Image Enhancement:** Based on primary object classification, this effective contribution plays a pivotal role in enabling visual clarity for images having single, dual, or multiple objects, shown in Figures 4 and 5. This is achieved by the support of the proposed classification model depicted in Section 2.2 and filtering techniques [37].**Classification Model**: The proposed model utilizes a deep learning technique. It classifies multiple objects as primary/non-primary. This model has two branches, which we term as image and object-specification branches. Both have the same purpose of extracting features from each and then concatenating before the linear layer is applied. This design was selected for the purpose of improving the model’s performance, specifically based on the suggested set of selective features.**Feature Selection and Dataset Preparation**: We utilized the common objects in the context (COCO) dataset as a base for our work. The SSD and depth estimation models were applied to the dataset so that the annotations could be processed and added to the dataset for better performance of the classification model. Furthermore, additional features were calculated, such as the area, size, and depth of objects, which are considered key factors in improving the classification model’s performance.

Section 1.1 provides a brief overview of the existing literature. In Section 1.2, the methodology is described with the details of the DNN-based [38,56] classification model construction. In Section 2.2, the proposed AI-assisted model is explained. The classification model network is also explained in detail. In Section 4, results of the proposed model are compared and justified by arguments. Section 5 provides a detailed discussion of existing research work to highlight a scope for improvement. Finally, a concrete conclusion is drawn in Section 6 based on the evaluation of the proposed model.

## 2. Proposed System

The proposed system image enhancement technique is tested on the COCO dataset. In this work, the proposed approach involves filtering the primary objects based on the engineered features. These feature enabled the modeling of a classification that is able to classify multiple objects as primary or non-primary.

### 2.1. Nomenclature

The nomenclature of this work is shown in Table 1. In this work, *i*, *j*, and *o_max_* are the number of images, number of objects in an image, and the number of maximum objects. In this research, *o_max_* is set to be 5. The coordinates of the top five objects are *x*_*o*,1_, *y*_*o*,1_, $x_{o,2}^{i},y_{o,2}^{i}$, $x_{o,c}^{i},y_{o,c}^{i}$, which refer to the top-left, bottom-right, and center coordinates of the bounding box, respectively. The other metrics are $c_{o}^{i},\;a_{o}^{i}\;d_{o}^{i}$, *p_o,score_*, and *D^i^*, which refer to the class, area, depth, and classification score of an object, and the depth estimations for the current image.

### 2.2. Network Architecture

In this work, a DNN is proposed to address the challenge of classifying image objects as primary or non-primary. This classification can then be used to filter out the unnecessary objects from the image to enhance the image required for image transmission to the artificial retina chip. This addition of AI-based preprocessing somehow overcomes the limitation of artificial retina chips. Further details of the proposed model and its interworking are as follows.

The proposed network architecture, as shown in Figure 3, addresses the multi-label classification task, which aims to predict whether the objects in image *i* are categorized as primary or non-primary. It takes two inputs: image *i*, a 2D grayscale image, and object detail information *o*, a 1D vector containing object information, such as position, size, and depth. The following model outputs five values that represent the primariness of these objects, i.e., $o_{max}$.

The architecture consists of two specialized branches for processing the distinct data types: image branch (i-branch) and object information branch (o-branch). Image *i* undergoes four 2D convolutional layers, followed by 2D global average pooling. This transforms the image into a 1D vector, while preserving spatial information. The object *o* information is processed through three 1D convolutional layers and 1D global average pooling, resulting in a 1D vector. The reason for choosing this specific NN is because of its ability to handle the extraction of features between the objects, which are highly complex in nature. The results of this model are evaluated and show that the top five, four, three, two, and one object(s) primary classification accuracy, precision, recall, specificity, and f1-score are satisfactory and lead to optimized image enhancements, shown in Figure 4.

Table 2 shows the metrics used in the network model. It can be visualized in Figure 3 that there are two branches, the i-branch and o-branch, in which both 2D and 1D convolutions are applied, respectively. Each of the convolution filter in_channels and out_channels are also listed, of which the shape of the output matrix is derived. Also, kernel and stride configurations are mentioned and are used in the network model. Global Average 2D and 1D are both applied after the convolutions, which have the output_size of (1,1) and 1. Then, finally, a dropout layer is applied to prevent overfitting, as there might be some zero-padded objects in some images. Then, finally, the 180 input_features and 5 output_features are used, resulting in five outputs over which a sigmoid is applied to scale the values between 0 and 1.

The i-branch takes a 256 × 256 image as an input, which is passed through four 2D convolution layers each followed by a layer except the last convolution layer. The size of the image adjusts to the shape of (15,128,128), (15,128,128), (30,64,64), (60,32,32), and (120,16,16), respectively, through the four convolution layers.

The 1D vectors of the two branches are concatenated into a single expansion vector, to which a dropout is applied, and then processed by a fully connected layer and a sigmoid activation function. The final output provides probabilities indicating whether each object in image *i* is primary or non-primary. This network architecture effectively combines image and object information, enabling multi-label classification, and addresses complex tasks involving diverse data types.

The overall proposed network architecture is modeled with the fewest possible processing components. Due to this, the proposed model can be called an optimal and effective model. The use of a single linear fully connected layer also indicates the fact that the feature design has been well worked out, leading to good results, shown in Section 4. In addition, the initial components not only extract the initial features, but also reduce the size of input to be fed to the linear fully connected layer. This reduction in input size and preservation of the information it contained in it contributes greatly to the performance of the proposed model. This is one of the reasons that a fully connected layer was sufficient for learning. Overall, the simple architecture is quite powerful and optimized at every level in optimizing the results with the network in mind.

## 3. Availability of Benchmark Dataset and Equipment

The dataset used in this work is downloaded from COCO benchmark’s website [47], which is widely used in computer vision research. The diversity involved object classes, Although other factors are also included in the dataset, such as light conditioning, which is another direction and needs separate research to be conducted. As the primary object classification goal is to easily identify the necessary objects required for optimized vision, this dataset serves that purpose and can be used for this type of research work.

In Table 3, the dataset used for experimentation is depicted. It is visible that the proposed model takes two types of inputs, i.e., a grayscale image and five object details associated with each image. The number of images it contains is 5000. It was downloaded from the COCO dataset’s official website [47]. Each image is a grayscale image with a size of 256 × 256. Each image has its top five object details associated in the form of a 5 × 10 matrix, in which 5 represents the number of objects and 10 refers to the number of details for an object *o*. Ten columns are depicted in the dataset table, which are the top-left coordinates $x_{o,1}^{i},y_{o,1}^{i}$, bottom-right coordinates $x_{o,2}^{i},y_{o,2}^{i}$ of the bound box, and central coordinate $x_{o,c}^{i},y_{o,c}^{i}$ derived from the top-left and bottom-right coordinates. The dataset columns associated with objects also include the class $c_{o}^{i}$, area $a_{o}^{i}$, depth $d_{o}^{i}$, and prediction score $p_{o,score}$. The column referring to the primariness $pr_{o}$ of an object is used as the target variable.

## 4. Experimental Results

This section provides the detailed explanation of setting up the experimental environment and description of the dataset. Following the evaluation results of the proposed model, a concrete conclusion is drawn at the end of this section.

### 4.1. Experimental Setup

In Table 4, a list of essential software applications is shown. All experiments are performed in the Windows subsystem for Linux (WSL) version 2. Anaconda version 22.9.0 is used to create a virtual environment [57] to avoid package and library conflicts. All the packages mentioned in the table are used for data preparation, loading, preprocessing, and visualization. The rest are used for model creation, training, analysis, and result generation.

Some of the libraries used in the dataset annotation are in PyTorch, which is explained in conjunction with Section 1.2, where it mentions the use of an SSD model for object detection and monocular depth estimation model for the distance of the objects from the camera. Based on both information types, further features are annotated based on calculations to enrich the selected feature set. These features played an important role in the learning of relationships and patterns between multiple objects.

### 4.2. Training

The dataset shown in Table 3 was partitioned to split the data into training, validation, and test datasets. The training, validation, and test data comprise 80%, 10%, and 10%, respectively. As the name implies, validation is performed along with training, and after achieving a satiable accuracy, test data were used to evaluate the performance of the model.

Other details about the training are shown in Table 5. The number of epochs is 400. The learning-rate is 0.0005. The batch-size for training is 64. The optimizer used for learning during training is the adaptive moment estimation weight (AdamW) with a binary cross-entropy (BCE) loss-function. The threshold set for classifying the object was set to the value of 0.5 initially during the training phase. Followed by training, with the help of derived metrics, the threshold value of 0.389 was found to be optimal, resulting in better accuracy.

### 4.3. Evaluation Results

This section provides a detailed evaluation of the performance of the proposed model in terms of the optimized image enhancements, the model training, validation, and testing the loss, accuracy, precision, recall, and specificity.

#### 4.3.1. Optimized Image Enhancements

In this subsection, the effect of edges and simulated phosphene are evaluated in comparison to the proposed enhanced images. In addition, the intensity profile analysis is made for each image such that the optimal performance results can be confirmed.

**Edges and Phosphene Comparisons**: In Figure 4, the results of several techniques applied to the original images can be visualized. The first three columns in the plot represent the original, its edge, and simulated phosphene. The next three columns in the plot represent the image with primary classified objects, its edge, and simulated phosphene. It is clearly evident that the objects are clearly identifiable as compared to the first three images in the plot. This is because unnecessary information is removed from the image with the support of the proposed multi-object classification model.The eight images in Figure 4 have either a single object or two objects, and the simulated phosphene results can be considered satisfactory in terms of vision clarity as compared to the phosphene applied over the original image.
Figure 4Different techniques for original vs. object-focused images, illustrating singular and dual object enhancements.
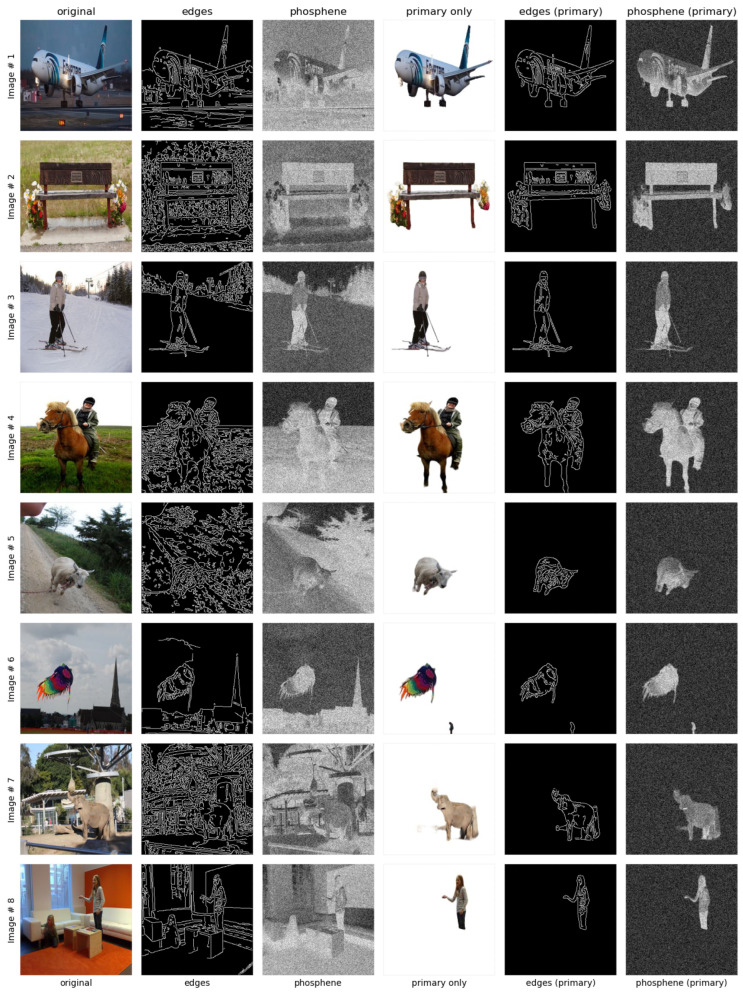

In Figure 5, it can also be visualized that there is a little ambiguity in identifying the overlapped objects, cat and television, in Images# 9, 10. But if the Images# (11–13) are considered for evaluation, even with the inclusion of multiple objects, the objects are easily identifiable, hence depicting the proposed approach’s optimized image enhancement.
Figure 5Different techniques for original vs. object-focused images, illustrating multi-object enhancements.
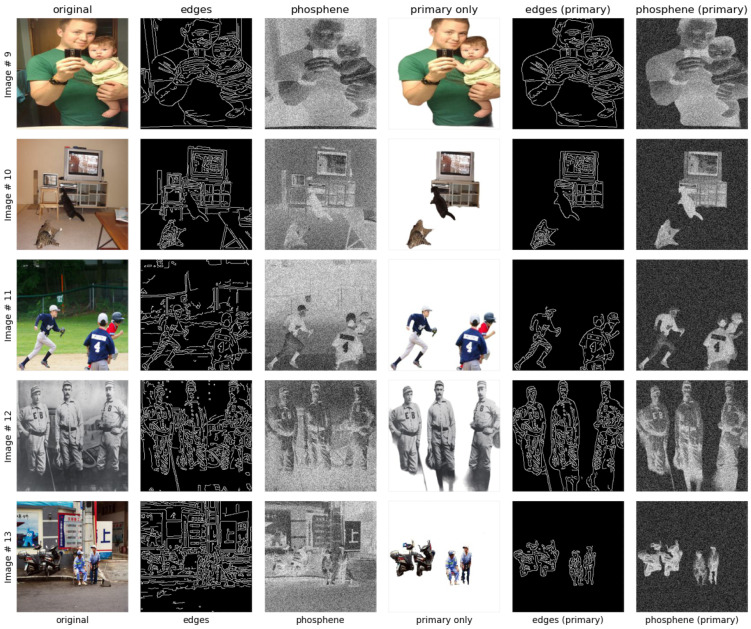

**Intensity Profile Analysis**: The intensity profile comparison between the original and proposed images is shown in Figure 6. The blue and orange lines represent the original and proposed enhanced image intensity profiles. This graph contains the intensity profiles for images containing one or two objects. It is clearly evident that the orange line, i.e., the proposed image’s intensity profiles, have an increased contrast and sharper peaks. This states the fact that the images are enhanced significantly in addition to the clarity in vision.
Figure 6Intensity profile analysis comparison between the original and enhanced images featuring singular and dual objects.
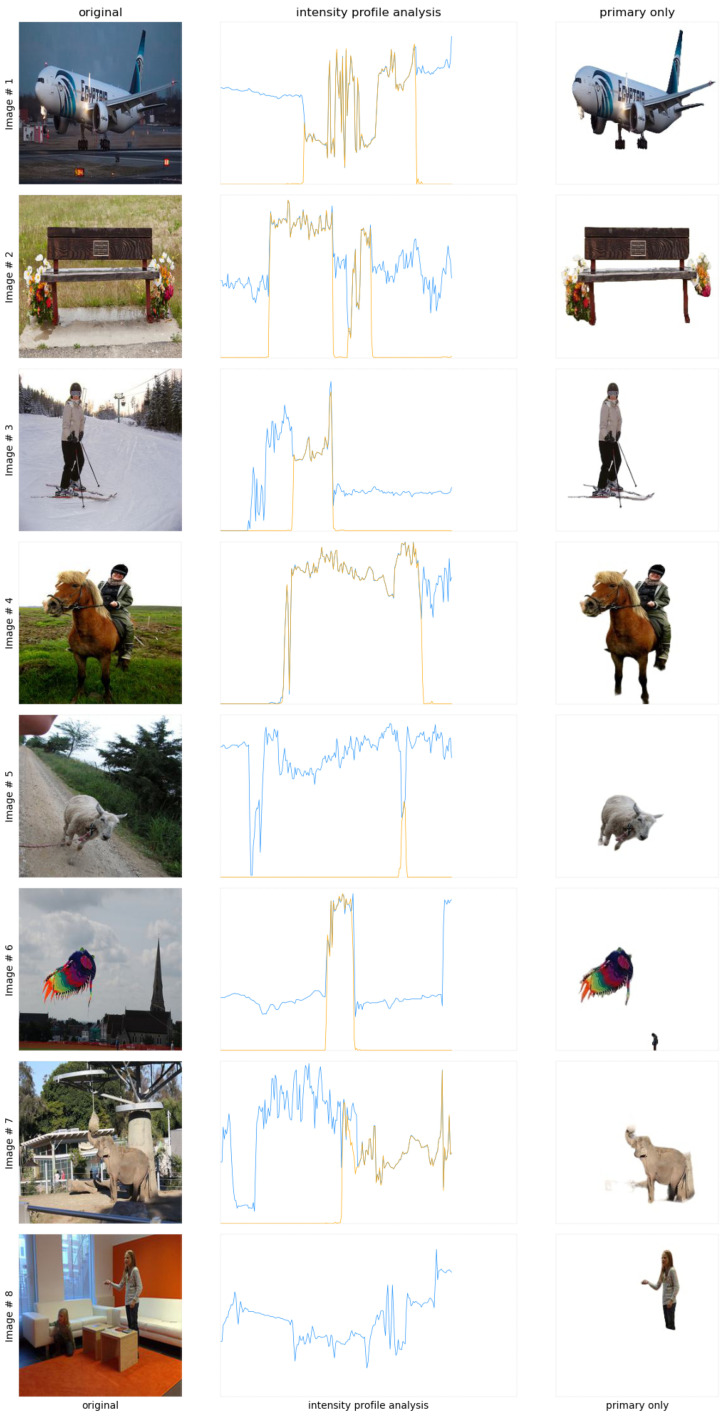
The second intensity profile comparison for images containing three or more objects is shown in Figure 7. It can be clearly analyzed that the proposed enhanced image intensity profiles show increased contrast and sharper peaks, which refers to the fact that the proposed system can cope with the challenges involved in multiple object scenarios. Referring to this, it can be stated that the images are enhanced significantly and kept along with the clarity in vision.

#### 4.3.2. Training and Validation Loss

Figure 8 shows the training and validation loss during the epoch. In the graph, the blue and pink colors represent the training and validation losses, respectively. The line graph shows that both losses are very low and very close to each other, which indicates the fact that the training results are quite good and there is no possibility of overfitting. It can be concluded that the proposed model gives very little loss while training, thus making it suitable. From this, the result is that the proposed model can be considered a good candidate for training in this specific classification problem.

#### 4.3.3. Validation Accuracy

Figure 9 represents the validation accuracy during the epochs. It is evident from the incline in the line graph that the accuracy is increasing as the epochs proceed, which reflects a fair enough accuracy, i.e., 86.2%. During the calculation of accuracy, the top five objects are considered to reflect a fair analysis for each object of the image. Detailed accuracy results for top ‘n’ objects are provided in later sections. It can be concluded from this result that the training along with the validation accuracy show a fair enough result, and hence, the proposed model can be considered a good candidate for this specific classification problem when considering a higher accuracy for all objects.

#### 4.3.4. Average Loss—Test

Figure 10 represents the average losses over the test data. The sky-blue, peach, green, red, and purple colors refer to the top one, two, three, four, and five objects. It can be depicted from the loss values that the top five objects’ results have the least value. The loss values for the top one, two, three, four, and five objects are 0.0062, 0.0045, 0.00385, 0.003, and 0.00257, respectively. Based on the loss values, it can be concluded that the loss values are low enough to refer to the fact that the proposed model can be stated as a good candidate for this specific classification problem. With this said, the top ‘n’ object analysis refers to a fair comparison of average losses.

It is evident that the accuracy decreases as the true positives (TP) for ‘n’ objects increases but the difference is negligible and, thus, provides a fair enough accuracy for the classification problem, i.e., 86.2%. The accuracies for the top one, two, three, four, and five objects are 96.4%, 91.8%, 86.8%, 85.2%, and 84.8%, respectively. This result refers to the fact that the proposed model is a reliable candidate for classifying the top five objects because it guarantees a higher accuracy. With this said, it can be concluded that the proposed model makes a fair analysis over the top ‘n’ objects with a higher accuracy and, hence, is a good candidate model for this specific classification problem.

#### 4.3.5. Accuracy—Test

Figure 11 represents the average accuracy over the test data. It is evident that the accuracy decreases as the TP ‘n’ objects increases but the difference is negligible and, thus, provides a fair enough accuracy for the classification problem, i.e., 84.8%. The accuracies for the top one, two, three, four, and five objects are 96.4%, 91.8%, 86.8%, 85.2%, and 84.8%, respectively. This result refers to the fact that the proposed model is a reliable candidate for classifying the top five objects because it guarantees a higher accuracy. With this said, it can be concluded that the proposed model makes a fair analysis over the top ‘n’ objects with a higher accuracy and, hence, is a good candidate model for this specific classification problem.

#### 4.3.6. Confusion Matrix

In Figure 12, the confusion matrix is depicted, which refers to the confusions made in the classification of the primary and non-primary objects of the image. The green, brown, blue, orange, and purple colors represent the fifth, fourth, third, second, and first object details. Each colored box is called a confusion matrix that represents each object. Each of the confusion matrices contain the TP, false positives (FP), false negatives (FN), and true negatives (TN), which are placed in each confusion matrix’s top-left, top-right, bottom-left, and bottom-right boxes, respectively. Against each confusion matrix, which represents each object’s statistics, it is evident that there are higher values of TP and TN as compared to the FP and FN, which refer to the fact that the accuracy is high for each object separately. In parallel, the lower values of FP and FN refer to the low error in the classifications made. Hence, the proposed model can be considered a good candidate for this specific classification problem.

#### 4.3.7. Accuracy, Precision, Recall, Specificity

Figure 13 depicts the accuracy, precision, recall, specificity, f1-score, Matthews correlation coefficient (MCC) rate, FP rate, and FN rate for all objects separately. Each comparison is for the top ‘n’ (five, four, three, two, and one) object(s), respectively.

#### 4.3.8. Precision Recall Area under Curve (PR-AUC)

Figure 14 represents the precision plotted against the recall values. The area under the curve can be seen to be very large for precision and recall, PR-AUC. As the precision reflects the accuracy of TP and minimizes FP, and the recall focuses on minimizing the FN value, the AUC can be increased only by proposing a fair enough threshold value. Based on the experiments performed, the optimal threshold value found was 0.3899, which produces the highest value of AUC for this experiment, i.e., 0.8751. Concluding the graph analysis, it can be stated that the proposed model is fair enough for the trade-off between precision and recall due to its higher value of PR-AUC.

All the results described in terms of evaluation suggest the use of the proposed classification model to support the object filtering process. It can be seen that the model performs fairly well in terms of training/validation loss and validation accuracy. Also, the average loss and accuracy for the top ‘n’ (five, four, three, two, and one) object(s) is also sufficient to model. Also, TP and TN are very high in value and FP and FN are very low in value. These confusion matrices are used to derive the results of precision, recall, specificity, F1-score, and MCC rate, which are found to be very high in value, which refers to the fact that the model is performing well. The FP and FN rates are also very low, referring to a low error in the classification.

Overall, the results show that the proposed classification model can significantly improve the vision of artificial retina patients. Due to the high classification accuracy, non-primary objects can be filtered out and an enhanced image can be transmitted to the retina chip. Transmitting enhanced images makes it easier to identify multiple objects and also allows for overcoming the challenge of artificial retina chip limitation.

## 5. Discussion

The following discussion provides ample information about the strengths and limitations of this research work. Based on the aforementioned limitations, future directions of this work are provided, showing potential for extension.

The dataset was annotated by a program developed by our team and several models (object-detection [58,59] and depth-estimation [60]) are used to retrieve information about objects and their depths. Using this information, other metrics are derived and used as the annotations, which include objects and their actual sizes, depths referring to distances, and classes of the objects. The information used in the annotation is very basic, based on which the primary and non-primary objects are labeled. It is important to mention that the labeling mechanism with the other information, such as such scene and context included in it, could have been much better, but it has no effect on the performance of the proposed model.

All the code was tested on the COCO image dataset. As a start, the model is trained to work well enough with the images, but a DNN is planned to be modeled for video scenarios, as the artificial retina patient is not treated by an image alone. Object tracking will be our second enhancement area, as multi-object tracking (MOT) [61] is a recent research challenge in the field of video processing.

It is also important to mention the strength of the model. It is a very lightweight model and requires less training time. During the experiments, we found some target areas where the model network could be enhanced and plan to implement it for video processing.

Results are generated purely based on top ‘n’ objects, which sufficiently justifies the assessment of the classification problem. The research and development of encoding techniques is also a part of the future plan. This will make the training process reliable.

Another addition to the annotation can be added to the dataset input, namely, the angle. This needs to be included due to the importance of perspective for something that can provide more insights into the dataset. The addition of this input may also enhance the labeling process and result in a more reliable annotation of the dataset.

Overall, the results were found to be satisfactory. During this journey, the research team discovered many interesting challenges to address in this domain, for example, adding angle and scene to dataset inputs, improving the labeling mechanism, improving dataset preprocessing, and adding video-processing support. Even without these enhancements, the results show a significant impact on artificial retina performance by filtering out unnecessary image objects with the help of an AI-assisted model [62].

## 6. Conclusions

In summary, this study emphasizes the importance of feature extraction for primary object classification in artificial retina vision. Accurate classification leads to the removal of unnecessary objects effectively, resulting in enhanced vision where multiple objects are easily identifiable. The vision clarity and classification model performance is improved. Handling complex scenarios involving multiple objects is enabled by our approach through the support of effective visual depictions, model accuracy, and performance metrics. Future research directions include the adaptation of this work for video processing, enhancing the dataset by including more selective features, and leveraging large language models to find more complex and rich features. Overall, this research brings innovation to artificial retina implants by enhancing the visual quality through AI-assisted technology.

## Figures and Tables

**Figure 1 sensors-24-02678-f001:**
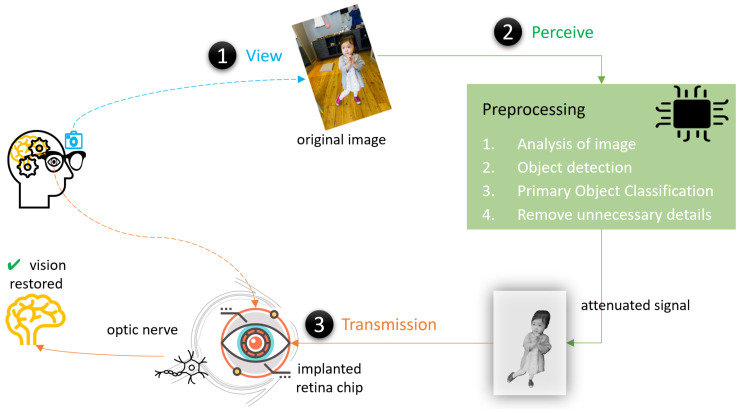
Components and process flow in AI-supported artificial retina.

**Figure 2 sensors-24-02678-f002:**
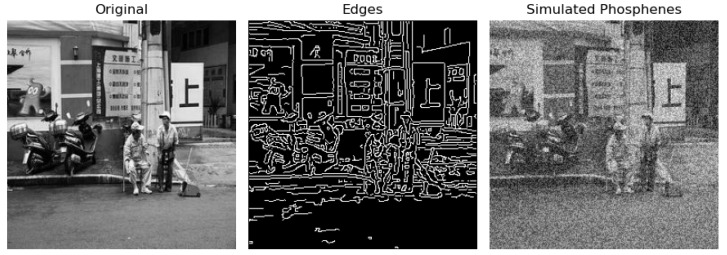
Edges and simulated phosphene.

**Figure 3 sensors-24-02678-f003:**
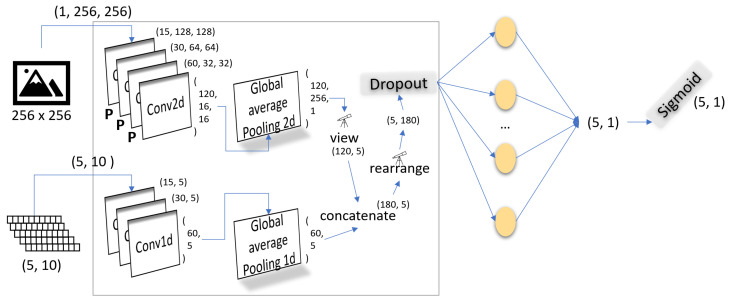
Network architecture.

**Figure 7 sensors-24-02678-f007:**
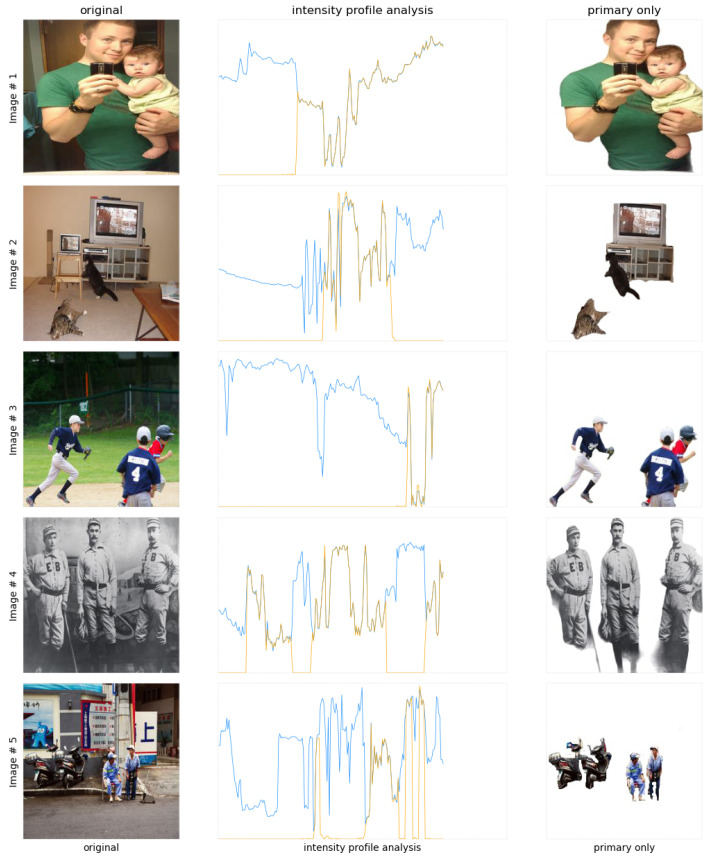
Intensity profile analysis comparison between the original and enhanced images featuring multiple objects.

**Figure 8 sensors-24-02678-f008:**
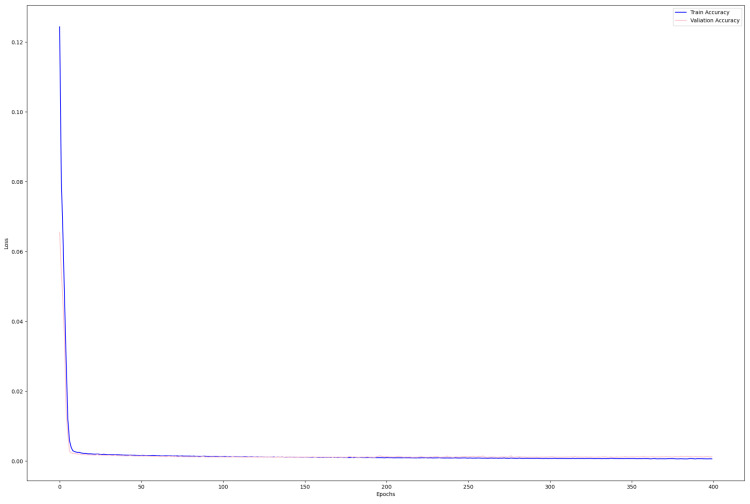
Training and validation loss.

**Figure 9 sensors-24-02678-f009:**
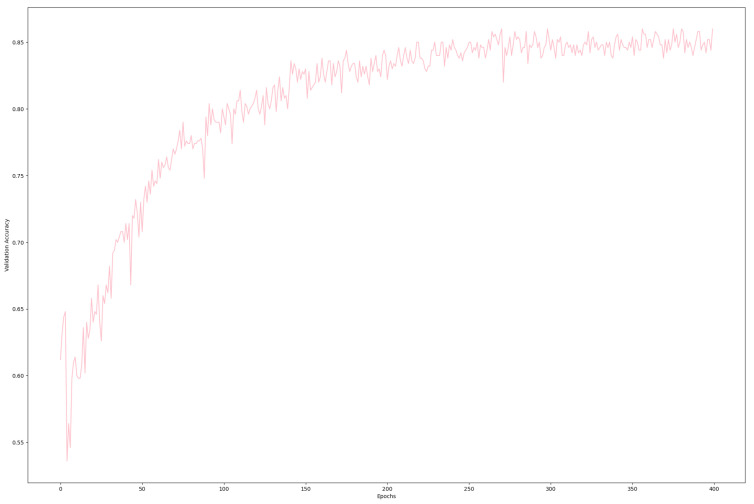
Validation accuracy.

**Figure 10 sensors-24-02678-f010:**
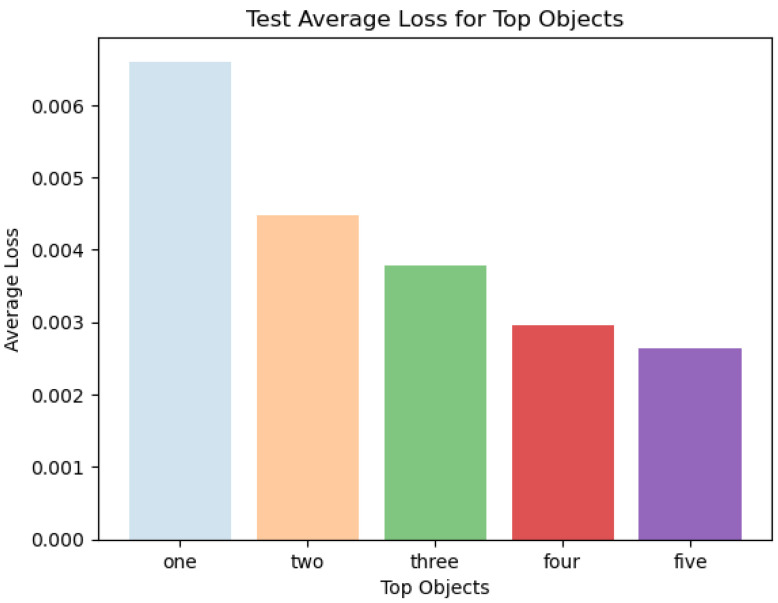
Average loss for top objects—test.

**Figure 11 sensors-24-02678-f011:**
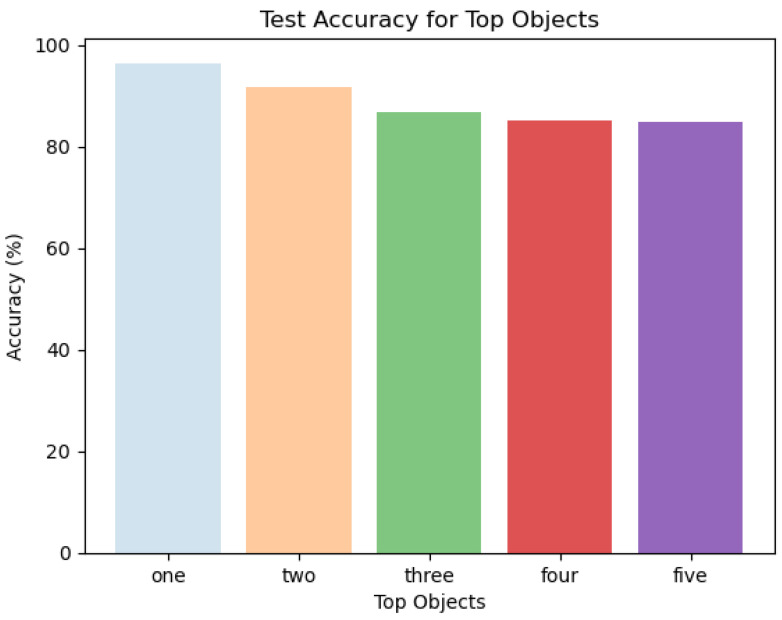
Accuracy for top objects—test.

**Figure 12 sensors-24-02678-f012:**
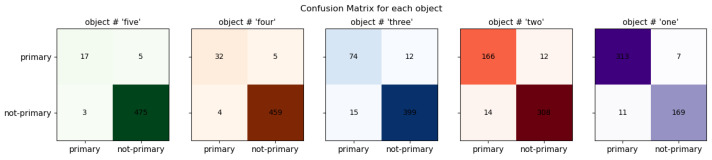
Confusion matrix.

**Figure 13 sensors-24-02678-f013:**
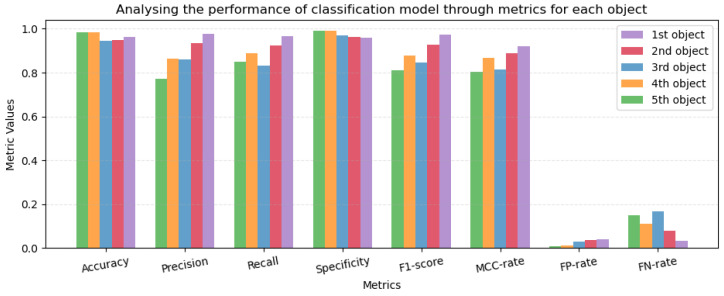
Accuracy, Precision, Recall, Specificity.

**Figure 14 sensors-24-02678-f014:**
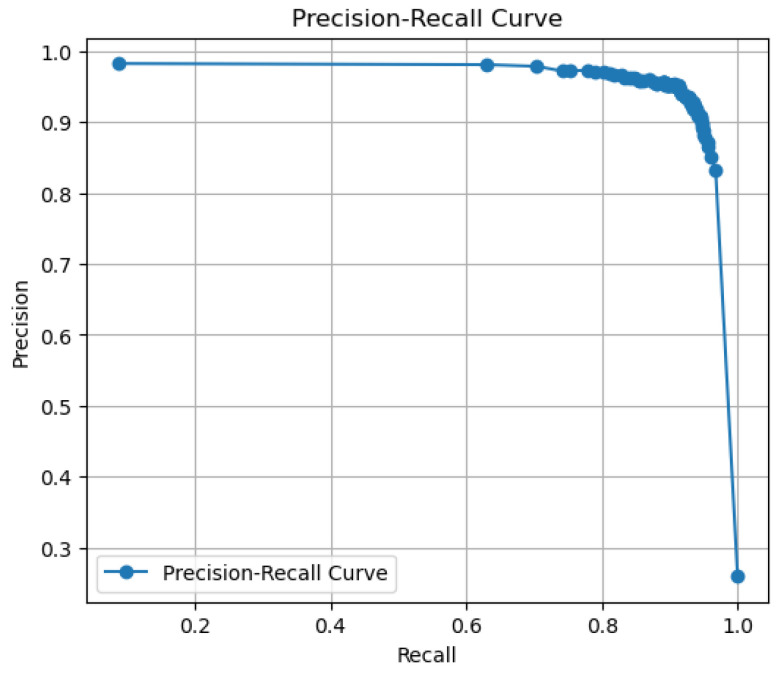
PR-AUC.

**Table 1 sensors-24-02678-t001:** Nomenclature of the classification model.

Symbol	Detail
*i*	number of images.
*o*	number of objects in an image.
*j*	instances of a class of object.
$o_{max}$	number of maximum objects.
$x_{o,1}^{i},y_{o,1}^{i}$	top-left coordinate of object *o*.
$x_{o,2}^{i},y_{o,2}^{i}$	bottom-right coordinate of object *o*.
$x_{o,c}^{i},y_{o,c}^{i}$	central-coordinate of object *o*.
$c_{o}^{i}$	class of object *o*.
$a_{o}^{i}$	area of object *o*.
$D^{i}$	depth estimations for image *i*.
$d_{o}^{i}$	depth of object *o*.
$p_{o,score}$	classification score of an object *o*.
$pr_{o}$	boolean representing primary/non-primary object *o*.

**Table 2 sensors-24-02678-t002:** Network architecture configurations.

Component	Layer	Metric	Value
i-branch	Convolutions	kernel_size	(3,3)
		stride	(1,1)
	Pools	kernel_size	(2,2)
		stride	(2,2)
	Conv2D(1)	in_channels	1
		out_channels	15
	Conv2D(2)	in_channels	15
		out_channels	30
	Conv2D(3)	in_channels	30
		out_channels	60
	Conv2D(4)	in_channels	60
		out_channels	120
	GlobalAvg2D	output_size	(1,1)
o-branch	Convolutions	kernel_size	3
		stride	1
	Pools	kernel_size	1
		stride	1
	Conv1D(1)	in_channels	1
		out_channels	15
	Conv1D(2)	in_channels	15
		out_channels	30
	Conv1D(3)	in_channels	30
		out_channels	60
	GlobalAvg1D	output_size	1
others	Dropout	zero_probability	0.5
	Fully connected linear layer	in_features	180
		out_features	5
	Final layer	activation_func	sigmoid

**Table 3 sensors-24-02678-t003:** Dataset.

Column	Detail
$D^{i}$	Depth estimations for image *i*, which is a 256 × 256 grayscale image.
$x_{o,1}^{i},y_{o,1}^{i}$	Top-left coordinate of the object *o* from the bound box.
$x_{o,2}^{i},y_{o,2}^{i}$	Bottom-right coordinate of object *o* from the bound box.
$x_{o,c}^{i},y_{o,c}^{i}$	Central coordinate of object *o* derived from $x_{o,1}^{i},y_{o,1}^{i}$, and $x_{o,2}^{i},y_{o,2}^{i}$.
$c_{o}^{i}$	Class representing type of the object *o*.
$a_{o}^{i}$	Area representing size of the object *o*.
$d_{o}^{i}$	This represents depth/distance of the object *o*.
$p_{o,score}$	Classification score representing the confidence score for an object *o*.
$pr_{o}$	This is a Boolean that represents whether the object *o* is primary/non-primary.

**Table 4 sensors-24-02678-t004:** Software applications.

Software/Library	Detail
WSL-2	1.2.5
conda	22.9.0
python	3.10.0
jupyter notebook	6.5.2
pytorch	2.0.1
pandas	1.5.3
numpy	1.24.3
PIL	9.4.0
cv2	4.8.0
transformers	4.30.2
matplotlib	3.7.1
Dataset (torch.utils.data)	2.0.1

**Table 5 sensors-24-02678-t005:** Configuration training settings.

Metric	Value	Metric	Value
epochs	1000	criterion	BCE Loss
learning-rate	0.0005	optimizer	AdamW
batch-size	128	threshold	0.5 (train)
			0.389 (test)

## Data Availability

The data can be found at http://images.cocodataset.org/zips/train2017.zip (accessed on 26 February 2024). We processed and annotated the images for the purpose of enabling desired experimentation.

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
