# Peer review of "Optimizing Image Enhancement: Feature Engineering for Improved Classification in AI-Assisted Artificial Retinas"

_sensors, 2024, doi:10.3390/s24092678_

Round 1

Reviewer 1 Report

Comments and Suggestions for Authors

A precise and comprehensive depiction of the architecture of the AI-assisted model is absent from the paper. Describe the precise algorithms and components that are utilised in visual restoration, elaborating on their functions and contributions.

Further specify how the model accommodates the distinct visual processing difficulties encountered by achromatic patients, including the lack of colour perception.

Analyse the potential constraints that may arise with regard to the reconstruction of authentic visual experiences for individuals who are achromatic at this time.

The present evaluation appears to be inadequate. Present both quantitative and qualitative findings to illustrate the efficacy of the AI-assisted model in enhancing the visual experience of individuals who are achromatic.

Incorporate information pertaining to metrics such as contrast sensitivity, colour discrimination, and visual acuity in order to evaluate the model's efficacy in reinstating diverse facets of vision.

It is recommended that a user study be undertaken to assess the subjective experience and level of contentment of a representative sample of achromatic patients with the AI-assisted artificial retina.

This essay will examine the constraints of the proposed model and evaluate its potential utility in addressing vision impairments other than achromatopsia.

Follow the "https://link.springer.com/article/10.1007/s11554-021-01125-8" paper. And represent the working model like this.

Propose prospective avenues for scientific inquiry and advancement in order to surmount these constraints and enhance the generalizability of the model.

Improve the structure and coherence of the manuscript. Avoid jargon and employ consistent terminology; organise the paper logically.  Enhance the calibre of figures and tables in order to impart information more effectively. Strictly examine the manuscript for typographical and grammatical errors.

Author Response

  English Korean Japanese Chinese (Simplified) Chinese (Traditional) Vietnamese Indonesian Thai German Russian Spanish Italian French Copy Support This Extension

Reviewer 2 Report

Comments and Suggestions for Authors

The paper is extremely interesting and of novelty. The authors described an AI based model that may be used in patients with artificial retina to provide color vision. Although presently, there are very few patients that were treated by artificial retina devices, advancements in this direction are extremely important.

As a minor issue, I would suggest if possible to add an example of how an image is perceived by an artificial retina device with or without the proposed model. 

Author Response

(The authors gave the same response as above.)

Reviewer 3 Report

Comments and Suggestions for Authors

The paper proposes an advanced artificial retina system equipped with an artificially intelligent model to detect and classify primary and non-primary objects in captured images. 

The motivation is to address the tissue damage caused by over-stimulation from the retina chip due to the abundance of objects in a scene. 

The proposed system employs a multi-label deep neural network classification model to down-select objects before transmission, reducing power stimulation requirements. 

The paper emphasizes the integration of AI in artificial retina technology to enhance image processing and improve the visual experience for the blind.

Evaluation:

1. Originality/Novelty:

The paper addresses a relevant and novel application of AI in the context of artificial retinas. 

The idea of using an AI-assisted model for object classification to reduce power stimulation is innovative. 

However, the novelty is somewhat diminished by the lack of a thorough discussion of related work, 

which is crucial for contextualizing the proposed approach within the existing literature.

2. Quality of Presentation:

The overall quality of presentation is acceptable but with notable issues. 

The absence of a related work section hampers the paper's clarity and completeness. 

Literature review should precede the methodology to provide a better understanding of the existing research landscape. 

There are English language issues, such as incomplete sentences and typos, that need correction to improve the overall readability.

3. Technical Contribution:

The technical contribution lies in the proposed AI-assisted model for object classification in artificial retinas. 

However, the lack of detailed explanation about the Graph Attention Network (GAT) and its relation to the proposed work is a significant drawback. 

Moreover, the paper misses an opportunity to discuss the broader field of attention networks, which is relevant to the proposed model. 

The paper also lacks an explanation of why the specific neural network architecture was chosen and how it compares to state-of-the-art models in this application.

Additional Comments:

- The paper does not sufficiently explain how the dataset used is representative of real-world scenarios, especially in the context of applying this technology to actual patients.

- The absence of a clear explanation for the choice of the neural network architecture and its comparison to existing state-of-the-art models is a significant gap.

- The technical details about the Graph Attention Network (GAT) and its role in the proposed model need to be elaborated for better understanding.

- The paper lacks a comprehensive discussion on attention networks, a crucial aspect in the proposed work.

- The evaluation metrics are mentioned, but a more detailed discussion on the experimental setup and comparison with existing methods would strengthen the technical contribution.

As a reviewer, I would recommend a "reject" for this paper. 

While the proposed idea of using an AI-assisted model for object classification in artificial retinas is interesting, 

several critical issues, including the lack of a related work section, insufficient explanation of the chosen neural network architecture, 

and unclear details about the Graph Attention Network (GAT), impact the overall quality and clarity of the paper. 

Additionally, the absence of a thorough discussion on attention networks and the dataset's representativeness to real-world scenarios further weaken the paper. 

Addressing these issues is crucial for the paper to make a significant contribution to the field and to be considered for publication.

Comments on the Quality of English Language

- English language issues, such as incomplete sentences and typos, need correction like for example:

 in line 325 there is an incomplete sentence "Each comparison"

 in line 69 "It this work" --> should be "In this work" 

Author Response

(The authors gave the same response as above.)

Round 2

Reviewer 1 Report

Comments and Suggestions for Authors

Describe how this document varies from these well-known earlier works and how it improves upon them. Your survey articles should include a study, and you should be sure to draw attention to any current gaps in knowledge.

Talk about some of the article's key elements, such as the paper's substantial contribution, future views, real-world/practical applicability, and the study's importance. 

In the introduction section, include some other research applications as literature. Some suggestions are:

A Review on Multi-Focus Image Fusion Techniques in Surveillance Applications for Image Quality Enhancement ; Revolutionizing Healthcare: The Application of Image Processing Techniques ;   Artificial Intelligence and Diabetes Mellitus: An Inside Look Through the Retina

It is recommended that the authors test their model using their own dataset in addition to the public dataset. It is necessary to compare the complexity. In this work, cross-validation is absent.

Use graphical analysis, such as intensity profile analysis or histogram analysis, for the optimum performance and confirmation of the results.

Rewrite conclusion. Add concluding, contribution points in it briefly. Abstract needs to rewritten with better summary of paper.

Author Response

(The authors gave the same response as above.)

Reviewer 3 Report

Comments and Suggestions for Authors

Considering my concerns raised in the first round of reviews

1. Originality/Novelty:

The authors have clarified their focus on the novel application of AI in the context of artificial retinas, specifically for feature extraction and object classification to reduce power stimulation. They have updated the manuscript to include a more detailed discussion of related work, addressing the initial concern about the novelty being somewhat diminished by the lack of a thorough discussion in this area. This enhancement strengthens the originality of the paper by better situating it within the existing literature and highlighting the innovative aspects of the proposed AI-assisted model.

2. Quality of Presentation:

The quality of presentation has seen notable improvements. The authors added a section on related work to the introduction, as per the suggestion, enhancing the paper's clarity and completeness. They also revised the manuscript to correct English language issues, including typos and incomplete sentences, thereby improving readability.

3. Technical Contribution:

The technical contribution of the proposed AI-assisted model for object classification in artificial retinas remains a significant aspect of the paper. In response to concerns about the lack of a detailed explanation about the Graph Attention Network (GAT) and its relation to the proposed work, the authors have clarified their focus on evaluating the impact of selected features over the designed classification model. They acknowledged the limitation regarding GAT and indicated ongoing and future work in this area. While some areas for further elaboration remain, such as the broader field of attention networks and the specific choice of neural network architecture, the authors have provided rationales for their current focus and direction.

4. Additional Comments:

The authors have addressed additional comments raised during the review, including the representativeness of the dataset used and the explanation for the choice of the neural network architecture. They have made revisions based on these comments, improving the manuscript accordingly.

Summary:

The final submission shows considerable improvement in addressing the concerns raised in the first round of review. The manuscript now better demonstrates its originality and situates itself within the existing literature, offers a clearer and more comprehensive presentation, and provides a more detailed technical contribution. While there are still areas for future exploration, such as a more in-depth discussion on attention networks and further comparisons with state-of-the-art models, the paper appears to have been significantly improved and may be closer to being ready for publication.

Author Response

(The authors gave the same response as above.)
